# An Armour Structure to Suppress the Brittle Failure of Ceramic Coatings

**DOI:** 10.3390/ma16144941

**Published:** 2023-07-11

**Authors:** Wei Liu, Fubing Bao, Yinning Zhang, Jinqing Wang, Xiaoyu Liang

**Affiliations:** 1School of Mechanical Engineering, Zhejiang Sci-Tech University, Hangzhou 310018, China; 2College of Metrology and Measurement Engineering, China Jiliang University, Hangzhou 310018, China

**Keywords:** ceramic coatings, armour structure, brittle failure, corrosion resistance

## Abstract

The brittle failure of ceramic coatings limits their application in many fields. To address this issue, a novel armoured ceramic coating was developed to suppress brittle failure. First, an interconnected frame microstructure was micromachined onto the surface of a mild steel substrate using a nanosecond laser. Subsequently, a polymer-derived ceramic slurry was sprayed and sintered to obtain an armoured ceramic coating. The laser-micromachined burr-like microstructure of the substrate facilitated adhesion between the coating and the substrate. The results of the mechanical properties test showed that the armoured coating could withstand more than 20 cycles of water-cooled thermal shock at 600 °C, and the peeling area of the armoured coating was approximately three times less than that of the unarmoured coating under a normal load of 1471 N. The laboratory and field corrosion test results indicated that at high temperatures, the corrosion resistance of the armoured coating was comparable with that of the unarmoured coating and was approximately 10 times higher than that of the uncoated sample. The proposed method will aid in suppressing the brittle failure of ceramic coatings and broaden their scope of application in different fields.

## 1. Introduction

Ceramic coatings are widely used as inorganic non-metallic materials in various applications, such as corrosion resistance [1,2,3,4], wear resistance [5,6,7], and thermal barrier coatings [8,9]. However, their low toughness is detrimental to their application in numerous fields [10]. For instance, at high temperatures, ceramic coatings are prone to cracks due to large temperature fluctuations or long-term operation in harsh environments, which reduces their efficacy in preventing corrosion at high temperatures. In addition, they are prone to impact during transportation, resulting in cracks, which, in turn, reduce their resistance to abrasion and corrosion. Therefore, reducing the generation of cracks is the key to improving the service life of ceramic coatings.

In modern materials science, the suppression of brittle failure is primarily achieved by microstructural alterations and the addition of secondary phases, which increase the toughness of ceramic coatings: transformation toughening [11], microcracking toughening [12,13], fibre/lamella bridge toughening [14], whisker toughening [15], frictional interlocking of grains [12], the addition of a piezoelectric secondary phase [16], and complex structure toughening. Although the toughness of ceramic coatings has been significantly improved through extensive studies conducted over the years, completely preventing brittle failure, especially in harsh environments, remains a formidable research challenge.

In recent years, polymer-derived ceramic (PDC) technology has been successfully applied in the processing of various ceramic coatings [17,18,19,20,21,22]. PDC is a technology that converts precursor polymers (such as polysiloxane [23], polysilazane [24], or polycarbosilane [25]) into an amorphous ceramic by pyrolysis between 500 and 1000 °C [26]. The drawback of PDC is the high shrinkage of the polymer during pyrolysis, which can be higher than 50% by volume [27]. Residual stresses caused by the high volume of shrinkage lead to the formation of cracks and pores or even delamination of the coatings. The addition of ceramic particles is usually used to reduce the volume shrinkage of the coating during sintering. However, the assistance of this method in improving the coating’s thickness and toughness is still limited.

In the case of unavoidable brittle failures of ceramic coatings, the phenomenon can be limited to local areas. On the basis of this concept, we constructed an armour microstructure similar to ‘mosaics’ in this study, i.e., the microstructure was an interconnected surface frame with ‘pockets’ that housed the ceramic coating, which was divided into independent small blocks. Although this armour structure can be applied to superhydrophobic surfaces [28,29], non-stick pans, etc., its application to ceramic coatings has not yet been reported. With this structure, the cracks generated by the ceramic coating can be controlled in the cells in harsh environments, such as large temperature fluctuations and the high local pressures induced by mechanical loads.

In order to obtain a connected framework structure on the metal’s surface, we adopted laser micromachining technology. This technology has been widely used for micromachining metal and ceramic surfaces to improve the tribological and mechanical properties [30,31,32,33]. Our work differed from studies in the current literature in that we combined micromachining of the metal surface and preparation of the coating, housing the coating in the armour structure. This not only increased the coating’s thickness, but also effectively suppressed the brittle failure of the coating.

In this study, we used laser micromachining to form an armour structure on the surface of TP347 metal material, which is commonly used as a surface material for boiler heating systems. The housed coating was prepared using organosilazane, which is widely used as a polymer-derived ceramic (PDC) [34]. We tested the thermal shock resistance, indentation, scratch resistance, and chloride corrosion resistance of the armour-structured coating, as well as performing a corrosion coupon test in a waste incinerator.

## 2. Materials and Methods

### 2.1. Materials

The coating suspension was prepared by following the methods reported by Wang [34] and Günthner [18]. In this experiment, organosilazane (KiONHTT1800, Clariant Advanced Materials) was used as the precursor, borosilicate glass powder (Glass 8470, Schott AG, Mainz, Germany) and ZrO_2_ (BYZ(Y-TZP)-2, Aladdin, Shanghai, China) were used as the fillers, and butyl acetate was used as the solvent to prepare the coating slurry. A commercial TP347 stainless steel was used as the substrate material.

### 2.2. Sample Preparation

The fabrication process of the armoured ceramic coating is illustrated in Figure 1a. First, a technical-grade ultraviolet nanosecond laser (TGNS-266; Nafei, Suzhou, China) was used to micromachine the interconnected frame structure on the surface of the metal substrate. Low-cost micromachining can be realized through the use of technical-grade lasers. Figure 1a shows the scale of the frame structures, and Figure 1b shows the metal specimen after laser micromachining. The silazane-based coating suspension was then sprayed onto the metal substrate using a spray gun (W71; Anest Iwata, Yokohama, Japan). Finally, the coated specimens were heat-treated in air at 700 °C for 1 h. Figure 1c shows the specimen after preparation of the coating.

### 2.3. Characterisation and Analyses

Thermal shock resistance tests (ASTM B571) were performed using a water-cooling method. The specimens prepared using the aforementioned process were immediately placed in a muffle furnace at 600 °C for 10 min, after which they were quickly removed and placed in a beaker filled with water. The specimens were then removed from the water and cooled to room temperature in air. This process was repeated, and the surface of the specimen after each thermal shock test was observed using a metallographic microscope (TS2500; Tusem, Shanghai, China).

We used the indentation method (ASTM C1327) to test the brittleness of the coating and assess the inhibition of crack growth in the armour structure. The indentation tests were conducted using a Vickers hardness tester (TH300; NDT-TIME, Beijing, China), which was used to apply a Vickers hardness load to the ceramic coating to generate indentation cracks. An HT-type indenter with an angle of 120° was used, and the normal load was 1471 N (Figure 2). 

The mechanical robustness of the armoured surface was demonstrated through sharp-object scratch tests, which were conducted using a cross-cut tester (DIN EN ISO 2409, MTV Messtechnik, Köln, Germany) as the indenter, as shown in Figure 3. The specimen was placed on a force sensor platform, which measured the force when the cross-cut tester scraped the specimen’s surface.

To assess the corrosion resistance of the armoured coating, laboratory-scale corrosion tests were conducted in a muffle furnace using KCl as the corrosion medium. The armoured coatings and samples with and without conventional coatings were tested. The samples covered with KCl were placed in the furnace at 600 °C for 168 h to corrode them. Subsequently, they were removed from the furnace and weighed using an electronic balance (ME104104E, Mettler Toledo, Zurich, Switzerland) with a sensitivity of 0.1 mg. The surfaces and cross-sections of the specimens were subjected to morphological analysis using scanning electron microscope (Supra 55S, Zeiss, Oberkochen, Germany) or a metallurgical microscope (TS2500, Tusem, China) before and after the abovementioned tests.

### 2.4. Corrosion Coupon Tests in a Waste Incinerator

To further verify the corrosion resistance of the armoured coating, we conducted coupon corrosion tests in a waste incinerator. The specimens were hung on the tubes of a high-temperature superheater using high-temperature corrosion-resistant nickel–chromium alloy wires. The temperature of the flue gas was approximately 540 °C, and the tests were performed for approximately 5 months.

## 3. Results and Discussion

### 3.1. Subsection Microstructure of the Armoured Coating

The interconnected laser-micromachined frame structure on the surface of the metal substrate is shown in Figure 4a,b. Owing to the inevitable thermal effect of the nanosecond-pulse laser micromachining process, melting, spraying, falling back, cooling, and solidification of the metal occurred on the side walls of the frame’s microstructure, forming a large number of burr-like microscale structures with an average height of approximately 10 μm (Figure 4b). These burr-like structures facilitated the adhesion of the coating to the substrate.

Figure 4c shows the surface morphologies of the sintered ceramic coatings, indicating a relatively dense coating surface, which resulted from the glass powders in the ceramic coating, which vitrified after the sintering and closely bonded to the ultrafine ceramic powder aggregates. Moreover, an interconnected frame structure was observed after the preparation of the coating. Figure 4d shows the cross-sectional morphologies of the sintered coatings; here, the armour structure is indicated by the yellow line. A high-magnification SEM image of the coating housed by the armour’s microstructure is shown in Figure 4e. The measured thickness of the coating was approximately 100 μm, which was comparable to the critical thickness obtained by Schütz [17] and Günthner [18] for coatings without armour structures. Due to the difference in the expansion coefficient between the ceramic coating and the metal substrate, a stress gradient appeared in the processes of sintering and thermal shock. Thicker coatings are prone to cracking, leading to the coating’s failure. We used the interconnected frame microstructure to divide the coating into independent small blocks. In this way, during the processes of sintering and thermal shock, the frame’s microstructure had very good buffer action and could reduce the generation of cracks, so that the thickness of the coating could be increased accordingly. When cracks occurred, they could also be confined to a separate frame to prevent widespread failure of the coating. The thickness of the coating measured in this study was correlated to the depth of the armour structure, indicating that the coating’s thickness could be increased by increasing the depth of the armour structure. An increase in the coating’s thickness could significantly improve the protective performance of the coating. The critical thicknesses of the coatings made by previous authors were small, mainly owing to the brittleness of the ceramic. 

As mentioned earlier, owing to the presence of the burr-like structures, the coating was closely inlaid on the substrate, thereby forming a metallic damascene structure, as indicated by the blue line in Figure 4e. However, holes may also have formed between the coating and the substrate, because the coating slurry or glass powder did not wet the burr-like structure during the spraying or sintering stages, respectively, as demonstrated by the red line in Figure 4e.

The presence of crystalline phases within the coatings was investigated by XRD (Figure 5). ZrO_2_ and ZnZrO_3_ were detected as the crystalline phases, which differed from the results obtained by Motz, where only ZrO_2_ was detected [18]. ZnZrO_3_ is a cocrystal formed at high temperatures by ZrO_2_ and ZnO in glass powder [35].

### 3.2. Thermal Shock Resistance Tests

As shown in Figure 6a, a few micropores were present on the surface of the coating without the armour structure before the thermal shock test. These were pinholes formed by the bubbles rising to the surface and breaking during the process of sintering the coating, which is similar to the glost firing process. After 20 thermal shock tests, many micropores appeared on the surface of the coating (Figure 6b), because during the sintering process, the glass powders melted and formed a glass layer with the increase in temperature. When the glass powders had completely melted, the gas generated in the coating was not removed and remained trapped. Then, during the cooling process, a fraction of the gas bubbles covered the coating’s surface and formed ‘glaze bubbles’. After the thermal shock, the ‘glaze bubble’ coating burst due to stress and formed pits. 

After 20 thermal shocks, the coating housed by the armour structure still adhered well, whereas that on the top walls of the frame microstructure peeled off under the thermal-shock-induced stress (Figure 6d). During the process of thermal shock, the frame microstructure had a very good buffer action and could reduce the generation of micropores.

In addition, under fluctuations in the external temperature, the peeling off of the coating in one block did not affect that in the other blocks, because the coating was divided into independent small blocks by the armour structure. This configuration prevented large-scale coating failure.

### 3.3. Indentation Tests

As shown in Figure 7a, the coating without an armour structure near the indenter peeled off due to brittleness during the indentation test. In contrast, no cracks or peeling were observed on the armoured coating under the same load in Figure 7b,c. We tested the peeling areas (including the indentation areas) of the coatings with and without armour structure under the same load, on the basis of the SEM micrographs, and the results are shown in Figure 7d. It can be seen that the peeling area of the armour coating (actually, this was the area of the indentation) was approximately three times less than that of the unarmoured coating. These results show that under external forces, the armour structure could effectively prevent the coating from peeling off and extension of the cracks.

### 3.4. Scratch Tests

Scratch tests were conducted to assess the ability of the armour structure to protect the coating under friction. The scratch load measured by the force sensor platform was approximately 7 N (Figure 8a), which led to the formation of irregular cracks/areas of spallation along the scratch’s path on the surface without the armoured coating (Figure 8b). By contrast, only the top wall of the frame’s microstructure of the armoured coating was significantly scratched by the same load (Figure 8c). In these tests, the tip size of the cross-cut tester was 0.05 mm, which was much smaller than the armour’s cavities (0.5 mm). Therefore, in the scratch tests, the force was excessively concentrated, resulting in scratches on the surface of the channels. The observations revealed that the armour structure could effectively protect the coating when scraped by a blunt object with a larger tip size. The scratch test results also revealed that the armour’s cavity size could be designed according to the tip size of the load object, so that the armour structure could protect the coating as much as possible.

### 3.5. Laboratory Corrosion Tests

Because the coatings attached to the top wall of the frame’s microstructure of the surface may peel off under fluctuations in the external temperature, armoured coatings subjected to thermal shock were used for the laboratory corrosion tests. Figure 9b shows the corrosion-induced weight gains of the samples with no coating, the conventional coating, and the armoured coating in KCl, which was used as the corrosion medium. Evidently, the corrosion-induced weight gain of the samples followed the order: non-coated sample > sample with the conventional coating > sample with the armoured coating (5.83, 0.41, and 0.61 mg·cm^−2^ respectively). This result demonstrates that the armoured coating can also play a protective role against corrosion. However, the corrosion resistance of the armour-coated sample was slightly lower than that of the sample with the conventional coating, because the top wall of the frame’s microstructure was not covered by the coating, and thus the substrate remained prone to corrosion from the top. Consequently, during the subsequent micromachining of the substrate, a frame microstructure with a narrow ridge width, similar to that used by Wang et al. [28], could be constructed, which ultimately reduced the area of the uncoated substrate.

Figure 9c,d shows the photographs and SEM micrographs of the corroded samples without a coating. Figure 9c shows the severe corrosion failure of the non-coated sample, and Figure 9d reveals that a loose corrosion layer formed on the surface of the sample, which was easy to flake off. In contrast, the conventional coating exhibited excellent corrosion resistance, although corrosion was observed at the sample’s edges. This edge corrosion occurred because of the absence of a coating on the sidewall of the sample, where the corrosion occurred and gradually eroded the coated area, resulting in the coating’s failure. Figure 9f presents the corroded, coated, and transition areas. The transition area shows the expansion of corrosion: the corrosion in the non-coated area expanded to the coated area and caused the coating to flake off. The micropores in the transition area were the internal pores of the coating, as indicated by the peeling off of the surface layer of the coating. Similarly, the armoured coating also showed excellent corrosion resistance. As evident from Figure 9h, no corrosion occurred in the housed coatings, whereas the top surfaces of the uncoated frame’s microstructures were corroded. Therefore, as previously mentioned, the top wall’s surface area in the frame’s microstructures should be minimised.

### 3.6. Field Corrosion Tests

Compared with the samples without a coating, the samples with conventional and armoured coatings showed sufficient protection against corrosion (Figure 10). As observed in the laboratory corrosion tests, the coating also peeled off at the edge of the samples because of the absence of coatings on the sidewalls of the samples.

## 4. Conclusions

In this study, a novel armoured coating was developed to suppress the brittle failure of ceramic coatings. The coating’s morphology, thermal shock resistance, brittleness properties, mechanical robustness, and anti-corrosion properties were studied, and the following conclusions were obtained. 

(1)The burr-like structural defects formed during the laser micromachining process promoted the formation of a metallic damascene structure between the coating and the substrate.(2)After 20 thermal shocks, no peeling of the armour coating was found, while many micropores appeared in the unarmoured coating.(3)The results of the indentation and scratch tests showed that the resistance to brittle failure, under an external force, was higher for the armoured coating than for the unarmoured coating. Under a normal load of 1471 N, the peeling area of the armour coating was approximately three times less than that of the unarmoured coating.(4)The results of the corrosion test showed that the corrosion-induced weight gain of the uncoated sample, the unarmoured coating, and the armoured coating was 5.83, 0.41, and 0.61 mg·cm^−2^ respectively. The armoured coating mostly retained its excellent corrosion resistance at high temperatures as well. The excellent anticorrosion performance of the armour coating was also verified in subsequent field corrosion tests.

The method presented in this study opens a route to suppress the brittle failure of ceramic coatings to broaden their applicability as environmental barriers in different fields.

## Figures and Tables

**Figure 1 materials-16-04941-f001:**
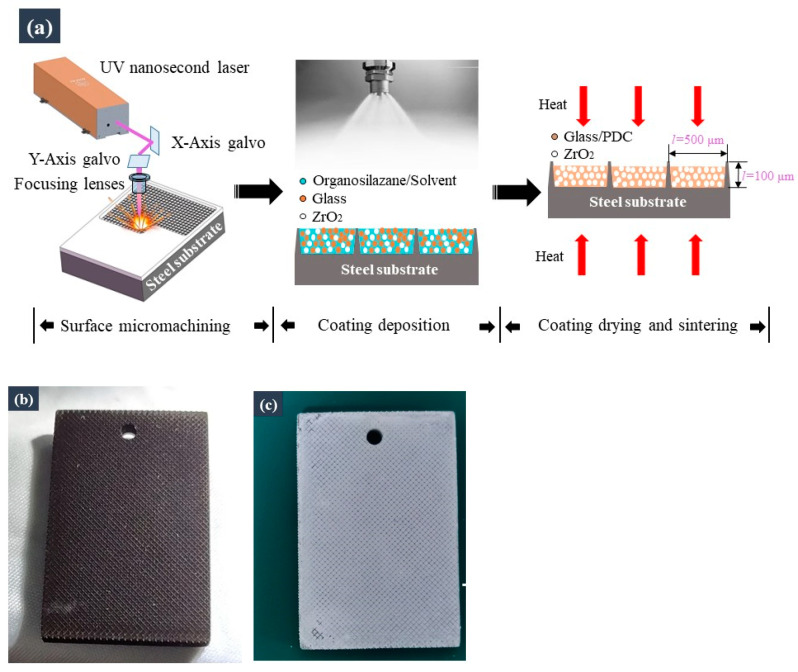
Schematic showing the fabrication of the armoured ceramic coating: (**a**) fabrication process; (**b**) the sample after laser micromachining; (**c**) the sample after preparation of the coating.

**Figure 2 materials-16-04941-f002:**
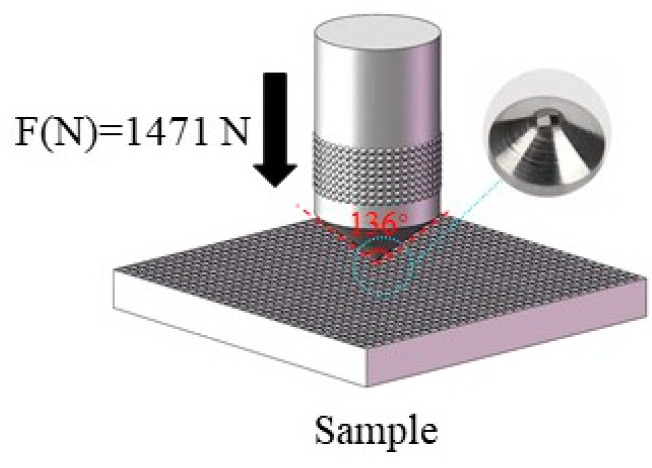
Schematic representation of the indentation test. A Vickers indenter was used under a normal load.

**Figure 3 materials-16-04941-f003:**
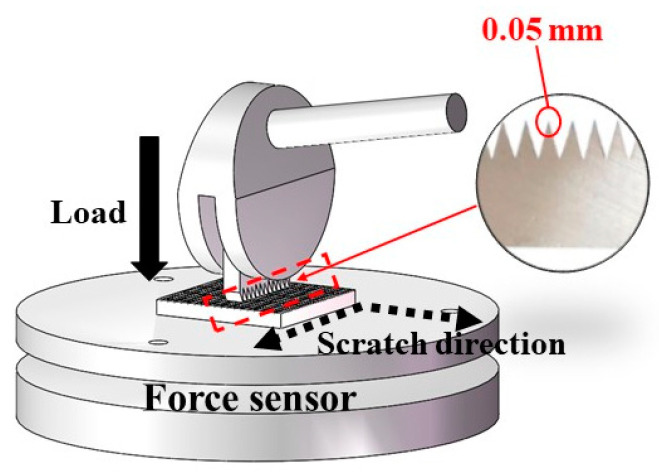
Schematic representation of the scratch test. A cross-cut tester was used as the indenter under a normal load.

**Figure 4 materials-16-04941-f004:**
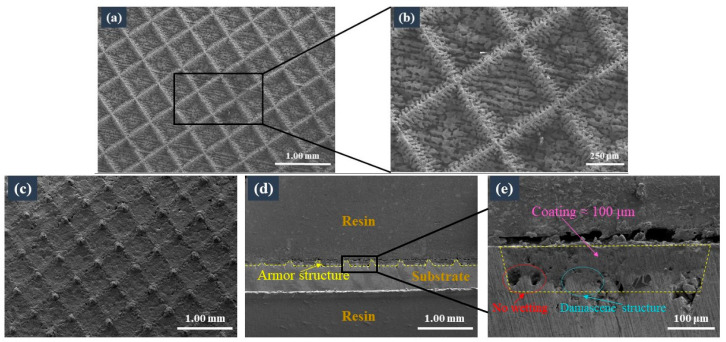
SEM micrographs of the armoured surface before and after preparation of the coating: (**a**) top surface before preparation of the coating; (**b**) top surface before preparation of the coating, at high magnification; (**c**) top surface after preparation of the coating; (**d**) cross-section after preparation of the coating; (**e**) cross-section after preparation of the coating, at high magnification.

**Figure 5 materials-16-04941-f005:**
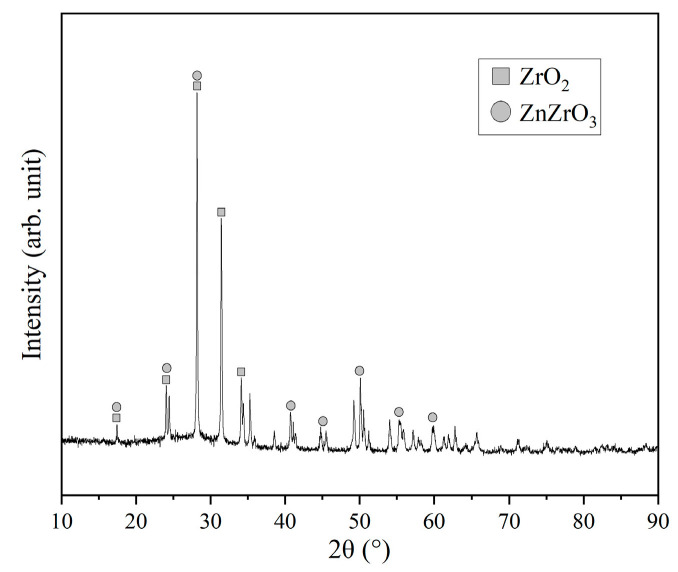
XRD pattern of the coated sample.

**Figure 6 materials-16-04941-f006:**
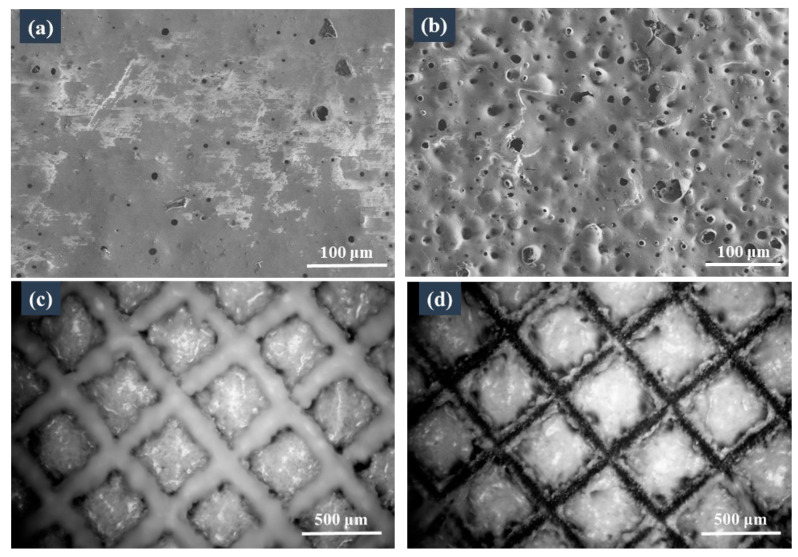
Micrographs of the samples before and after 20 thermal shock tests. SEM micrographs of the coating without an armour structure (**a**) before and (**b**) after the thermal shock tests; optical micrographs of the coating with an armour structure (**c**) before and (**d**) after the thermal shock tests.

**Figure 7 materials-16-04941-f007:**
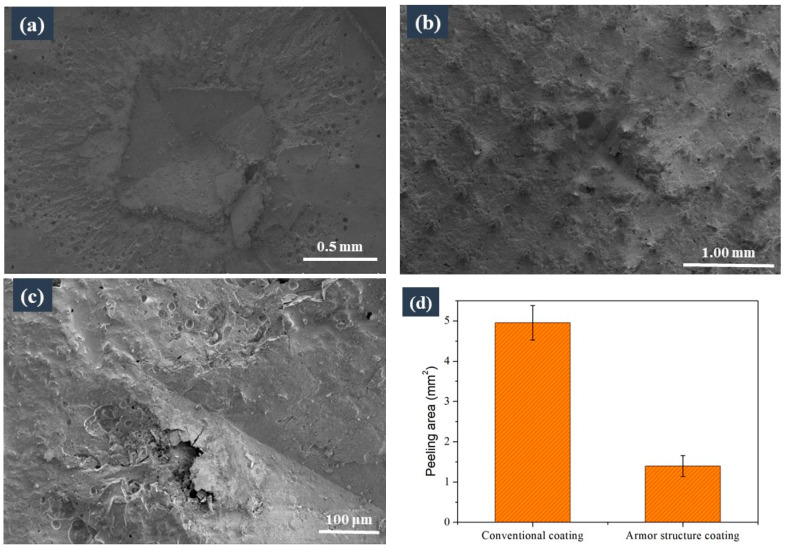
Comparison of the surface brittleness of the two samples based on the indentation method: the coating (**a**) without and (**b**) with the armour structure after the indentation tests; (**c**) high-magnification image of the coating with the armour structure after it was subjected to the indentation test; (**d**) peeling areas of the coatings with and without the armour structure.

**Figure 8 materials-16-04941-f008:**
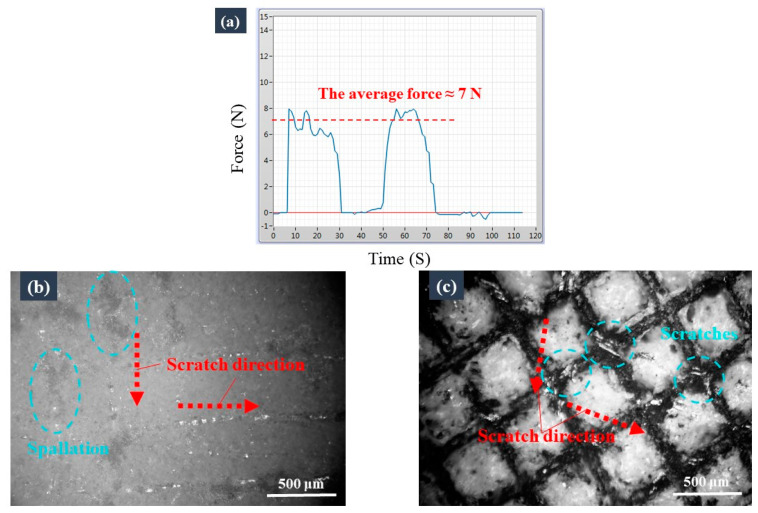
Comparison of the mechanical robustness of the two samples based on the results of the scratch test: (**a**) the scratch load measured by the force sensor platform; optical micrographs of the coatings (**b**) without and (**c**) with the armour structure after they were subjected to the indentation tests.

**Figure 9 materials-16-04941-f009:**
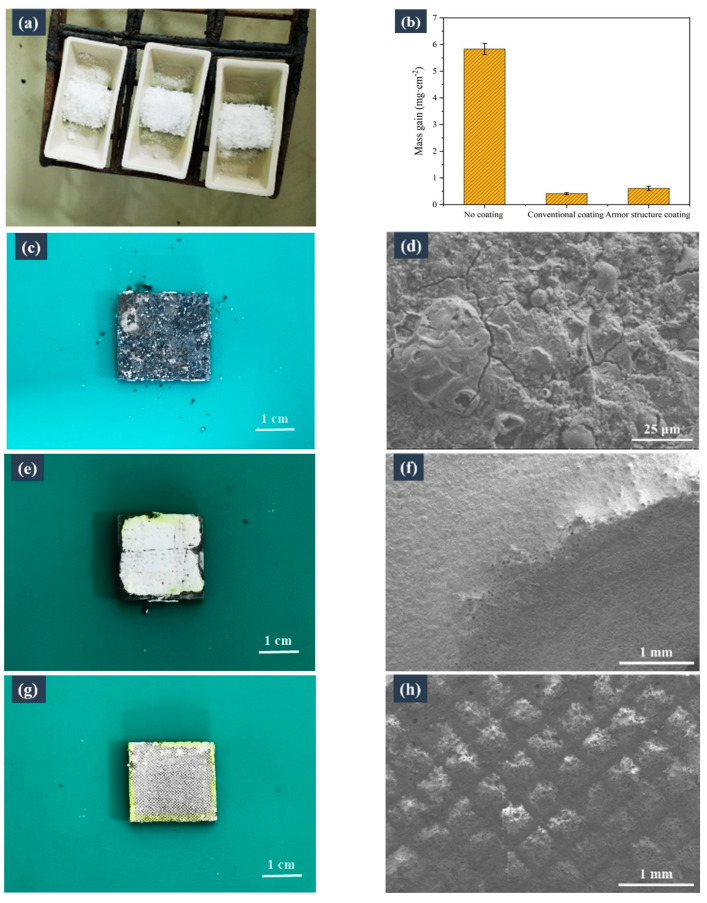
Comparison of the corrosion resistance of the samples subjected to laboratory-scale corrosion tests. (**a**) Corrosion coupons prior to exposure to high temperatures. (**b**) Corrosion-induced weight gains of the samples with no coating, the conventional coating, and the armoured coating in KCl (corrosion medium). (**c**,**e**,**g**) Photographs and (**d**,**f**,**h**) SEM micrographs of the sample surfaces with no coating, the conventional coating, and the armoured coating, obtained after the corrosion tests.

**Figure 10 materials-16-04941-f010:**
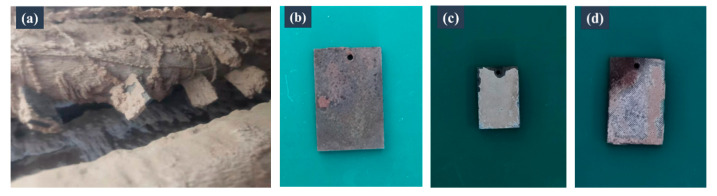
Comparison of the corrosion resistance of the samples based on the results of the field corrosion test. (**a**) Photograph of the samples subjected to the field corrosion test. Photographs of the samples with (**b**) no coating, (**c**) the conventional coating, and (**d**) the armoured coating after the field corrosion tests.

## Data Availability

Some or all data that support the findings of this study are available from the corresponding author upon reasonable request.

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
