# Peer review of "An Armour Structure to Suppress the Brittle Failure of Ceramic Coatings"

_materials, 2023, doi:10.3390/ma16144941_

Round 1

Reviewer 1 Report

1.      This paper discusses feasibility studies on armor structure to suppress brittle failure of ceramic coatings.

2.      In the abstract it is mentioned that the anti spalling performance of the armoured coating is superior to that of conventional coatings. The findings of work need to be presented in quantitative terms rather than expressing the same qualitatively.

3.      In the Introduction section proper research gaps need to be identified and accordingly the objectives of the study to be stated. Further novelty of the work to be stated clearly.

4.      In Section 2.2 Sample preparation images of the samples prepared need to be shown.

5.      ASTM standards used for testing the samples such as indentation method, mechanical robustness, corrosion resistance etc also to be included.

6.      In the results and discussion, it is mentioned the critical thickness of the coatings made by the previous authors were small, mainly owing to the brittleness of the ceramic. How the present thickness achieved (higher than the previous works) could overcome the brittleness issues?

7.      Results and discussion section need in depth analysis of the results (Thermal shock resistance tests, Indentation tests, Scratch tests) shown indicating co-relation to already published literature. Figure 6 (b) is not clear. For figures 6 (d, f, h) appropriate scale shall be presented.

8.      Conclusion section shall be revised to highlight the important findings of the work carried out and to be expressed in quantitative terms.

9.      Reference section to be revised involving recent papers from the literature can be added to make it more relevant for the researches working on the similar topic.

10.  Uncertainty error analysis for all the measurements needs to be presented.

11.  Manuscript needs careful correction for typographical errors. 

Moderate editing of English language can be done to improve the quality of the manuscript.

Reviewer 2 Report

In this manuscript, the authors prepared silazane based coatings on steel substrates to suppress the brittle failure of the ceramic coatings. This manuscript topic is suitable for publication in Materials Journal. Before publication, some of my comments need to be addressed. My comments are below.

1.      The author prepared coatings using low viscosity liquid polysilazane-based coating resin, which is commercially available. In the introduction, the author should discuss literature reported on coatings prepared through polysilazane resin. It is well-known in terms of polymer derived ceramics, and this must be discussed in the introduction part. I found some literatures in the website (https://d-nb.info/1256173215/34, file:///C:/Users/kamalan.mosas/Downloads/coatings-13-00537-v2.pdf ) . Please revise and compare your results with the already reported values. Also highlight the novelty of the work in the introduction part.

2.      Thermal shock resistance- why did the author choose 600°C for testing though the author uses 700°C for sintering process.

3.      What are X and Y axis description in Figure 5b.

4.      Why the author did not show interest in analyzing hardness and scratch resistance after corrosion testing. It seems all the coating was worn out after corrosion testing and only the coating portion available within the micromachining part.

5.      Conclusion should be brief. Please avoid general sentences.

6.      To confirm the effectiveness of the coating, impedance spectroscopy before and after corrosion testing may be performed.

7.      Overall, the study results are not encouraging. Justify your outcome.

Reviewer 3 Report

The authors introduced an armoured ceramic coating to address the issue of brittle failure in ceramic coatings. Using a nanosecond laser, the approach involved micromachining an interconnected frame microstructure on a mild steel substrate. A polymer-derived ceramic slurry was sprayed and sintered to create the armoured ceramic coating. The paper is interesting, but the novelty is not clear. Below is the feedback. 

1- the introduction is short and should address the literature in ceramics coatings processing and armours application also the processing of ceramic polymers is essential ie. : 10.1002/adem.200800158. so more references are required in the above area. 

2- the topic of ceramic coating on micromachining surfaces is not new and there are several papers in this area so what is the novelty of your work? examples: 10.1115/1.4048639; 10.3788/CJL201340.0803001; 

10.2351/1.5062344;  10.1016/j.optlastec.2013.05.007;  10.3788/CJL202148.1402011;  10.1016/j.ceramint.2019.03.197;  978-160560387-2;  10.1016/j.surfcoat.2014.09.021;  10.4416/JCST2017-00080. the experimental section:  the indentation and scratch method should be moved to the method section and results discussed in the results section.  the aim is to use it in armour applications then impact testing should be provided.  Additional SEM analysis and XRD should be carried out, identifying the elements and phases of the sintered parts.   

it is ok, but some sections could be improved 

Round 2

Reviewer 2 Report

Thank you for your revision.